# Facilitating the propagation of spiking activity in feedforward networks by including feedback

**Hedyeh Rezaei**[1], **Ad Aertsen**[2], **Arvind Kumar**[2,3]☯ *, **Alireza Valizadeh**[1,4]☯ *

**1** Department of Physics, Institute for Advanced Studies in Basic Sciences (IASBS), Zanjan, Iran, **2** Faculty of Biology, and Bernstein Center Freiburg, University of Freiburg, Freiburg, Germany, **3** Dept. of Computational Science and Technology, School of Computer Science and Communication, KTH Royal Institute of Technology, Stockholm, Sweden, **4** School of Cognitive Sciences, Institute for Research in Fundamental Sciences (IPM), Niavaran, Tehran, Iran

☯ These authors contributed equally to this work.
* arvkumar@kth.se (AK); valizade@iasbs.ac.ir (AV)

**Data Availability Statement:** All relevant data are within the manuscript and its Supporting Information files. The simulation code is available on github: https://github.com/arvkumar/Communication-Through-Resonance.

## Abstract

Transient oscillations in network activity upon sensory stimulation have been reported in different sensory areas of the brain. These evoked oscillations are the generic response of networks of excitatory and inhibitory neurons (*EI*-networks) to a transient external input. Recently, it has been shown that this resonance property of *EI*-networks can be exploited for communication in modular neuronal networks by enabling the transmission of sequences of synchronous spike volleys ('pulse packets'), despite the sparse and weak connectivity between the modules. The condition for successful transmission is that the pulse packet (PP) intervals match the period of the modules' resonance frequency. Hence, the mechanism was termed *communication through resonance (CTR)*. This mechanism has three severe constraints, though. First, it needs periodic trains of PPs, whereas single PPs fail to propagate. Second, the inter-PP interval needs to match the network resonance. Third, transmission is very slow, because in each module, the network resonance needs to build up over multiple oscillation cycles. Here, we show that, by adding appropriate feedback connections to the network, the CTR mechanism can be improved and the aforementioned constraints relaxed. Specifically, we show that adding feedback connections between two upstream modules, called the resonance pair, in an otherwise feedforward modular network can support successful propagation of a single PP throughout the entire network. The key condition for successful transmission is that the sum of the forward and backward delays in the resonance pair matches the resonance frequency of the network modules. The transmission is much faster, by more than a factor of two, than in the original CTR mechanism. Moreover, it distinctly lowers the threshold for successful communication by synchronous spiking in modular networks of weakly coupled networks. Thus, our results suggest a new functional role of bidirectional connectivity for the communication in cortical area networks.

**Funding:** This work was funded by the German Federal Ministry of Education and Research (BMBF) grant number 01GQ0830 (to AA), The Carl Zeiss Foundation (to AA), State of Baden-Württemberg through bwHPC and the German Research Foundation (DFG) through grant no INST 39/963-1 FUGG (to AA), The Ministry of Science, Research, and Technology (MSRT) of Iran grant (to HR), The Iran Saramadan Elmi Federation (ISEF) grant (to HR), Cognitive Sciences and Technologies Council of Iran, research grant No 832 (to AV), and Swedish Research Council's research project grant and StratNeuro (to AK). These funders had no role in study design, data collection and analysis, decision to publish, or preparation of the manuscript.

**Competing interests:** The authors have declared that no competing interests exist.

## Author summary

The cortex is organized as a modular system, with the modules (cortical areas) communicating via weak long-range connections. It has been suggested that the intrinsic resonance properties of population activities in these areas might contribute to enabling successful communication. A module's intrinsic resonance appears in the damped oscillatory response to an incoming spike volley, enabling successful communication during the peaks of the oscillation. Such communication can be exploited in feedforward networks, provided the participating networks have similar resonance frequencies. This, however, is not necessarily true for cortical networks. Moreover, the communication is slow, as it takes several oscillation cycles to build up the response in the downstream network. Also, only periodic trains of spikes volleys (and not single volleys) with matching intervals can propagate. Here, we present a novel mechanism that alleviates these shortcomings and enables propagation of synchronous spiking across weakly connected networks with not necessarily identical resonance frequencies. In this framework, an individual spike volley can propagate by local amplification through reverberation in a loop between two successive networks, connected by feedforward and feedback connections: the resonance pair. This overcomes the need for activity build-up in downstream networks, causing the volley to propagate distinctly faster and more reliably.

## Introduction

Anatomical differences and functional specialization of different brain regions suggest that the brain is organized as a highly modular system. This modularity can be observed in the neocortex at multiple spatial scales, ranging from inter-areal connectivity [1, 2] to inter- and intra-layer connectivity within a single cortical column [3–6]. A modular design indeed provides numerous benefits, not only making the system scalable, but also rendering it with robustness to structural perturbations [7].

To exploit the modularity of the brain, it is however, necessary that neuronal spiking activity from one specialized network can be reliably transmitted to another network and that the downstream network is able to read the incoming activity [8, 9]. Therefore, understanding how spiking activity is reliably propagated from one brain region to another is crucial for understanding the functional organization and information processing in the brain.

Different brain modules, irrespective of their spatial scale (inter-areal or inter-layer), are interconnected by convergent-divergent connections. Typically, the connectivity between any two modular networks is sparse, and synapses are weak [10]. Over the last decade, the problem of reliably transmitting spiking activity via weak and sparse connections has attracted much attention from experimentalists and theoreticians alike [8, 11–18]. If the inter-module networks under study exclusively include feedforward connections, the only way to overcome the problem of transmission with weak synapses is to provide more efficient signals by synchronizing the spike signals to be transmitted [19–21]. Neuronal signals in this case are considered as volleys of spikes (*pulse packets*) which can be quantified by the number of spikes in the volley ($\alpha = 50\text{--}100$ spikes) and their temporal dispersion ($\sigma \approx 1\text{--}10$ ms), measuring the degree of synchronization of the spiking activity in the volley [20, 22]. Several studies have demonstrated that the downstream effect of a pulse packet depends both on $\alpha$ and $\sigma$ (see [8] for a review). Note that a pulse packet by itself does not carry any information; rather, the information resides in the combination of neurons participating in the spike volley, both in the sender and receiver networks [23].

Convergent-divergent connectivity motif can generate and amplify spiking synchrony by virtue of shared inputs [19, 20, 23, 24]. When inputs are sufficiently synchronous, the transmission speed is very high and governed only by synaptic delays. However, it has been shown that this mechanism requires relatively dense connectivity and/or highly synchronous inputs [8]. These two requirements are inconsistent with available experimental data on both the neuronal connectivity and activity across cortical areas.

But cortical networks are not strictly feedforward, and recurrent and feedback connections are prevalent in the central nervous system [4, 5]. Network activity dynamics determined by recurrent connectivity have a strong effect on neuronal response properties. For instance, network oscillations modulate the neurons' spiking threshold in a periodic fashion. If two networks oscillate at the same frequency and phase (coherent oscillations), the transient decrease in the effective spiking threshold of neurons in the downstream network coincides with the transient increase of the spiking activity of the sending network, facilitating the transmission of spiking activity [9, 11, 12, 14, 17, 18]. Thus, *communication through coherence* (*CTC*) not only provides the means to communicate from one network to another, but it also provides the means to control the communication, because only networks with an appropriate phase synchrony with the sender network can tune in to the spiking activity they receive. Thus, CTC requires that spontaneous coherent oscillations exist between the sender and receiver networks before the onset of stimulus-evoked activity to be transmitted and that the coherence remains stable, despite continuous shifts in frequency and phase of the oscillations [25]. However, mechanisms underlying such coherent oscillations have so far remained obscure (however, see [26, 27]).

Recently, Hahn and colleagues proposed another mechanism that does not require coherent spontaneous oscillations in the sender and receiver networks before the arrival of activity that needs to be propagated. Instead, it is based on the evoked oscillations following the impact of a stimulus [17]. For a wide range of biologically plausible neuron and network parameters, excitatory-inhibitory networks (*EI*-networks) show features of network resonance. In this regime, the baseline activity of the network itself is not oscillatory, but when perturbed with a transient input, the network responds with a damped oscillation. When stimulated with a periodic external input with the appropriate frequency, within a few oscillation cycles the network starts to oscillate at its intrinsic oscillation frequency.

Thus, even a weak periodic input, provided it has the right frequency, exposes the network resonance and creates oscillations in the receiver network which would not exhibit oscillations otherwise. Network oscillations created through this resonance phenomenon periodically lower the spiking threshold of neurons in the receiver network, allowing for a gradual build-up, over several oscillation cycles, for enabling the transmission of the incoming activity. Therefore, this mechanism was termed *communication through resonance* (*CTR*) [17]. Because oscillations only arise upon appropriate stimulation of the downstream network, the oscillations in the sender and receiver networks are automatically locked in an appropriate phase for transmission and, hence, facilitate the transmission of the spiking activity involved in the stimulation. Thus, the *CTR* mechanism resolves a fundamental problem of the *CTC* hypothesis: how to obtain and, even more so, maintain phase synchrony between the network oscillations. Yet, at the same time it creates new problems: First, it precludes the transmission of individual pulse packets and, second, because the periodic stimulation activity needs to be amplified by build-up over multiple oscillation cycles, communication through resonance is prohibitively slow. Finally, it is not known how the inter-pulse interval of the external signal can be matched to the period of the evoked oscillations of the modules.

Here, we report the results of an investigation how the transmission of spiking activity in a feedforward network (FFN), based on the CTR mechanism, can be improved by adding

appropriate feedback connections. To this end, we studied the possibility of transmitting a single pulse packet in an FFN of *EI*-networks in which the first two layers of *EI*-networks were bidirectionally connected via weak and sparse excitatory synapses. We refer to these two bidirectionally coupled *EI*-networks as the *resonance pair*. We found that adding such a resonance pair to an otherwise feedforward modular network enabled fast transmission (in only two oscillation cycles) of a single pulse packet through a built-in CTR mechanism, provided the sum of the feedforward and feedback delays between the resonance pair matches the period of the resonance of the *EI*-networks. In the FFN with a resonance pair, the incoming single pulse packet initiated a periodic pulse packet train with appropriate timing (determined by the resonance frequency of the *EI*-networks), which was reliably transmitted through the remainder of the layered network of *EI*-networks. We found that the build-up of the network resonance was much faster in networks with a resonance pair: embedding a single resonance pair in a feedforward network increased the speed of CTR-based transmission by a factor of 2. Using numerical simulations, we identified conditions (strength, number and delay of the bidirectional connections) that ensured a stable transmission of the activity, without destabilizing the activity dynamics within the individual *EI*-networks in the layered network. We hypothesize that, since bidirectional connections between cortical areas are quite ubiquitous (e.g. [28–33]), such bidirectionally connected areas may provide good broadcasters of information in the brain at intermediate and large scales.

## Methods

### Neuron and synapse model

Neurons were modeled as leaky integrate-and-fire (LIF) neurons. The sub-threshold dynamics of the neuron's membrane potential were described by:

$$C_m \dot{V}_m = -G_{leak}[V_m(t) - V_{reset}] + I_{syn}(t) \tag{1}$$

where $V_m$ denotes the membrane potential, $C_m$ the membrane capacitance, $G_{leak}$ the membrane leak conductance, and $I_{syn}$ the total synaptic input current. When the membrane voltage reached the threshold of $V_{th} = -54$ mV, a spike was emitted and the potential was reset and clamped to $V_{reset} = -70$ mV for a refractory period ($\tau_{ref} = 2$ ms). To avoid a transient network synchrony at the beginning of the simulation, the initial membrane voltage of neurons was drawn from a normal distribution (mean: −70; standard deviation: 3 mV). The neuron model parameters are listed in Table 1.

Synaptic inputs were introduced by a transient change in the synaptic conductance $G_{syn}$:

$$I_{syn}(t) = G_{syn}(t)[V_m(t) - E_{syn}] \tag{2}$$

in which $E_{syn}$ denotes the synaptic reversal potential. Conductance changes were modeled as

**Table 1. Neuron parameters.**

| Name | Value | Description |
| --- | --- | --- |
| $C_m$ | 250 $pF$ | Membrane capacitance |
| $G_{leak}$ | 16.67 $nS$ | Membrane leak conductance |
| $V_{th}$ | -54 $mV$ | Spiking threshold |
| $V_{reset}$ | -70 $mV$ | Reset potential |
| $\tau_{ref}$ | 2 $ms$ | Refractory time period |

alpha functions:

$$G_{syn}(t) = \frac{t}{\tau_{syn}} exp(-\frac{t}{\tau_{syn}})$$                    (3)

where $\tau_{syn}$ is the synaptic time constant. The synapse model parameters are listed in Table 2. Here we considered weak synapses [10], such that in the default case, the co-activation of ~50 excitatory presynaptic neurons was required to elicit a spike in the postsynaptic neuron. When we systematically varied the excitatory feedforward or feedback strength (cf. Figs 4 and 7), the numbers of co-activated presynaptic neurons required to elicit a spike were different. Synaptic transmission delays were set to 1.5 ms for within-layer connections; whereas inter-layer transmission delays were systematically varied as one of the key parameters in our study (as mentioned in the corresponding Figure captions).

## Network connectivity

The network consisted of 10 layers, each one comprising 200 excitatory and 50 inhibitory neurons in the form of an *EI*-network (Fig 1). The connectivity within the layers (*EI*-networks) was chosen to be random with a fixed connection probability of 0.2 for all types of connections. For the inter-layer connectivity, we assumed that only the excitatory neurons from one layer *EI*-network projected to the excitatory neurons in the following layer *EI*-network. From each layer, 70 randomly selected neurons projected to the next layer with connection probability of = 0.2. Thus, each neuron in a layer received on average 40 excitatory inputs from neurons within the layer network and 14 excitatory inputs from neurons in the preceding layer network. Synapses from a neuron onto itself were excluded, but multiple synapses between neurons were allowed. Inter-layer excitatory connections were set to be as strong as within-layer excitatory to excitatory connections. In the case of feedforward networks (FFN), all connections between adjacent layers were unidirectional. In the case of the resonance pair network (RPN), we introduced feedback excitatory connections between the first two layers of the FFN. We took care that individual neurons were not bidirectionally connected. Strength, probability and delay of the feedback and feedforward connections were systematically varied to identify conditions for resonance between the two layers (Figs 7 and 8). Further details of the model network parameters are listed in Table 3.

**Table 2. Synapse parameters.**

| Name | Value | Description |
| --- | --- | --- |
| $\tau_{exc}$ | 1 *ms* | Rise time of excitatory synaptic conductance |
| $\tau_{inh}$ | 1 *ms* | Rise time of inhibitory synaptic conductance |
| $E_{syn}^{exc}$ | 0 *mV* | Reversal potential of excitatory synapses |
| $E_{syn}^{inh}$ | -80 *mV* | Reversal potential of inhibitory synapses |
| $J_{ee}$ | 0.33 *mV* | Exc. to exc. synaptic strength measured at -70 mV |
| $J_{ei}$ | 1.5 *mV* | Exc. to inh. synaptic strength measured at -70 mV |
| $J_{ie}$ | -6.2 *mV* | Inh. to exc. synaptic strength measured at -54 mV |
| $J_{ii}$ | -12.0 *mV* | Inh. to inh. synaptic strength measured at -54 mV |
| $J_{pe}$ | 0.25 *mV* | Connection strength: Input Poisson spike train to exc. pop. |
| $J_{pi}$ | 0.4 *mV* | Connection strength: Input Poisson spike train to inh. pop. |
| $J_{pp}$ | 0.33 *mV* | Connection strength: Pulse packet to P neurons in first layer |
| $d_{within-layer}$ | 1.5 *ms* | Transmission delay within layer |
| $d_{inter-layer}$ | 25–28 *ms* | Range of total resonance delay between layers |

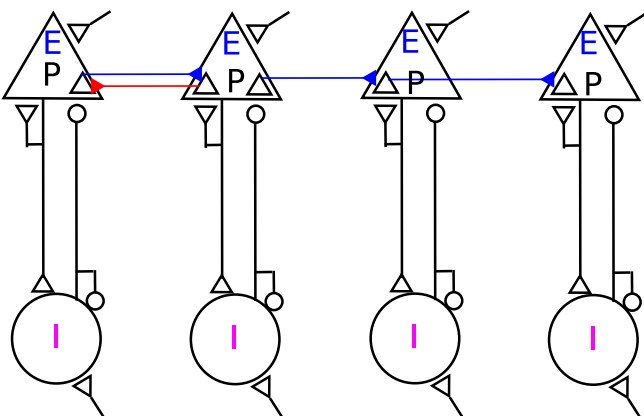

**Fig 1. Schematic representation of a feedforward network with a resonance pair.** 200 excitatory neurons in each layer (E), including 70 projecting neurons (P), and 50 inhibitory neurons (I) have random homogeneous sparse recurrent connections. Ten layers are connected sparsely through *EE* connections, indicated by blue arrows, in a feedforward manner. The red arrow from layer 2 to 1 indicates sparse random feedback connections from the second to the first layer *EI*-network, for which we used the term **resonance pair**.

**Table 3. Network parameters.**

| Name | Value | Description |
|---|---|---|
| $N_{exc}$ | 200 | Size of excitatory population per layer network |
| $N_{inh}$ | 50 | Size of inhibitory population per layer network |
| $N_{proj}$ | 70 | Number of projecting neurons per layer network |
| $\epsilon_{within\text{-}layer}$ | 0.2 | Connection probability within-layer network |
| $\epsilon_{inter\text{-}layer}$ | 0.2 | Connection probability between layer networks |

## External background input

Each excitatory neuron in each layer *EI*-network was driven by 8, 000 independent Poisson excitatory spike trains, each with a mean rate of 1 spike/s. Each inhibitory neuron in each layer *EI*-network was driven by 6, 400 independent Poisson excitatory spike trains, at the same mean rate. In Fig 7a and 7b, the rate of the Poisson input to the *E*-neuron population was systematically varied, and for the *I*-neuron population the rate was adjusted accordingly, to keep the difference between the mean input rates to *E*- and *I*-neurons, 1, 600 spikes/s, constant. Network connectivity, synaptic strength and external input were tuned such that each individual layer *EI*-network operated in an asynchronous-irregular regime [34, 35] in the absence of pulse-packet like inputs. This also meant that the network was operating in an inhibition-dominated regime [34, 35]. However, the network operating point was close enough to the oscillatory regime, such that a single pulse packet stimulus could elicit weak damped oscillations, which we exploited to create resonance by external stimulation.

## Synchronous input

The synchronous input stimulus was a single pulse packet, injected into the projecting neurons in the first layer network. It consisted of a fixed number of spikes ($\alpha$), distributed randomly around the packet's arrival time ($t_n$). The time of individual spikes were drawn independently from a Gaussian distribution centered around $t_n$, with a standard deviation of $\sigma = 2$ ms. In Figs 2d, 2e and 6, the external input for the FFN was a periodic train of pulse packets with inter-

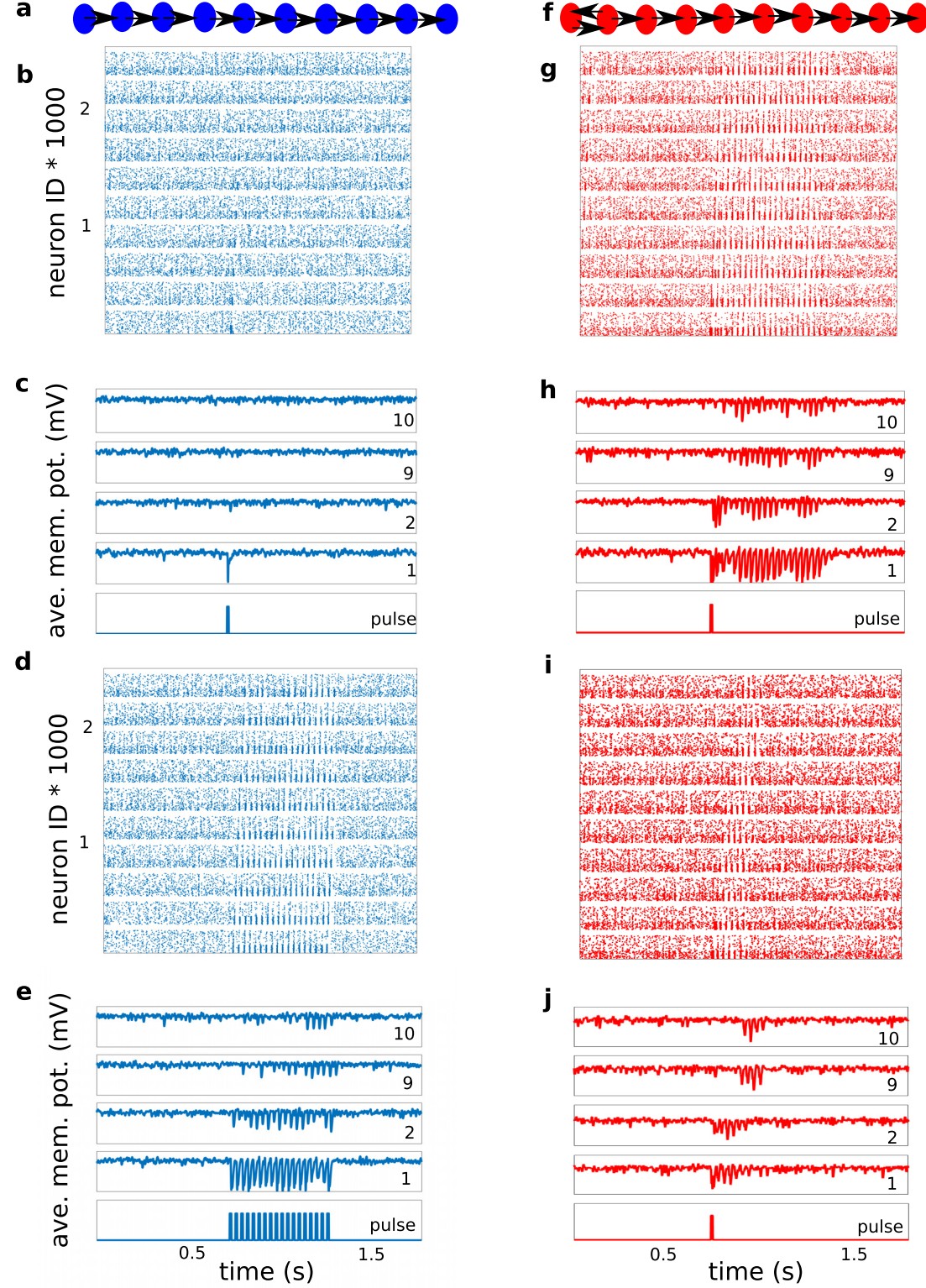

**Fig 2. Comparison of the propagation of synchronous spiking in a feedforward network (FFN) and in a resonance pair network (RPN).** The FFN failed to propagate a single pulse packet **(a-c)**, whereas it did propagate a periodic train of pulse packets with the appropriate time interval between successive pulse packets **(d, e)**. By contrast, the RPN, when stimulated with a single pulse packet, was able to propagate it successfully, provided that the inter-layer delay of the resonance pair matched the resonance period of the *EI*-networks involved **(f-j)**. Panels **(i)** and **(j)** are similar to what was shown in panels **(d)** and **(e)**, the

only difference being the increased Poisson input rate to the inhibitory neurons in panels **(i)** and **(j)** in order to decrease the number of stimulus-evoked oscillation cycles in the RPN. In the simulation experiment shown in panels **(g-j)**, the loop transmission delay, defined as the sum of the forward and feedback transmission delays, was equal to the period of the pulse packet train in **(d)** and **(e)**. The network structure for each column is plotted schematically in panels **(a, f)**, the corresponding raster plots are shown in panels **(b, d, g, i)** for each stimulus condition. The average membrane potentials of the first two and last two layers in each of three simulation experiments are shown in panels **(c, e, h, j)**, marked with layer numbers in each window, with the injected pulse packet shown in the bottom trace. Red color is used for the RPN, and blue for the FFN. Inter-pulse interval in panels **(d)** and **(e)** was 25 ms and the forward and backward delays in panels **(g-j)** were equal to 12.5 ms.

packet intervals of 25 ms. In Fig 6c, $\alpha$ was a control parameter and was varied systematically. In all remaining cases we used $\alpha = 20$. In most cases, we stimulated the RPN with a single pulse packet, but in some cases we tested how a train of periodic pulse packets propagated through the RPN (S3a Fig). In that case, we systematically varied the interval between subsequent pulse packets. Finally, to study whether the RPN can also transmit irregular trains of pulse packets, we jittered the pulses by a small amount ($dt$), while keeping the mean inter-pulse packet interval constant. The amount of jitter ($dt$) was quantified in terms of the network's resonance period ($T$) and was randomly varied between 0 and up to $T/2$ (S3b Fig).

## Data analysis

**Pairwise correlations.** To estimate pairwise correlations, we divided the time into bins of size $\Delta t = 5$ ms, and transformed population spike trains to spike count vectors $y_i(t)$, using a rectangular kernel. The pairwise Pearson correlation coefficients were calculated as:

$$r_{ij} = \frac{\mathbb{E}[(y_i(t) - \bar{y}_i(t))(y_j(t) - \bar{y}_j(t))]}{\sigma_i \sigma_j} \tag{4}$$

where $\mathbb{E}$ denotes the expectation value, $\sigma$ the standard deviation, and barred values denote the means of variables. We averaged the $r_{ij}$ over all pairs within a layer network to obtain the average synchrony within the layer. Correlation coefficients were computed from simulations with a duration of 20 sec and were averaged over 20 trials.

**Population fano factor.** To classify the population activity based on synchrony in the background activity, we measured the population Fano factor (*pFF*) [36]. To this end, we used spike count vectors ($y(t)$) of all excitatory neurons in a layer network and defined the *pFF* as:

$$pFF = \frac{VAR[y(t)]}{MEAN[y(t)]} \tag{5}$$

This normalized variance of the population activity is related to synchrony of the population activity because the population variance is the sum of individual variances of the neurons and their co-variances. Unlike pairwise correlations, FF is a measure which takes into account not only pairwise correlations, but higher order correlations as well [35].

**Network frequency and spectral entropy.** The network frequency is defined as the peak frequency of the Fourier transform of the spike count vectors $Y(f)$. To differentiate between asynchronous, aperiodic and oscillatory states of the two resonance pair networks, we measured the spectral entropy [37] of one of the two *EI*-networks involved. We first calculated the power spectrum $S(f) = |Y(f)|^2$ and defined:

$$P(f) = \frac{S(f)}{\sum_m S(m)} \tag{6}$$

Because $P(f)$ has unit area, we treated it as if it were a probability density and estimated its entropy. Our reasoning was that if a signal is periodic, all its power will be concentrated in a single frequency, resulting in a zero entropy. By contrast, when the signal power is uniformly distributed over all frequencies, the entropy will assume a maximal value. Given that we estimated the spectrum for a fixed number of frequency data points, we needed to normalize the entropy according to the number of frequency bins. Because here we took the normalized power spectrum as a proxy for the probability density, we refer to this measure as spectral entropy. The normalized spectral entropy is then defined as [37]:

$$H = -\frac{\sum_{f=1}^{N} P(f) \log_2 P(f)}{\log_2 N}, \tag{7}$$

where $N$ is the number of frequency data points. The denominator, $\log_2 N$ is the maximal spectral entropy, that is, the spectral entropy of white noise. Low entropy indicates temporal order of the population activity, i.e., an oscillatory state, whereas large values of $H$ indicate an asynchronous state.

**Signal-to-Noise Ratio.** To distinguish successful propagations of single pulse packets from failed propagations, we estimated the *Signal-to-Noise Ratio (SNR)*, measuring the variance of the spike count vector in the tenth (final) layer network upon pulse packet injection into the first layer network, normalized by its variance during ongoing network activity:

$$SNR = \frac{VAR[y^{stim}(t)]}{VAR[y^{ongoing}(t)]} \tag{8}$$

## Simulation tools

Network simulations were performed using the simulation tool NEST (http://www.nest-initiative.org) [38, 39], interfaced with PyNest. The differential equations were integrated using fourth-order Runga-Kutta with a time step of 0.1 ms. The simulation code is available for download from https://github.com/arvkumar/Communication-Through-Resonance.

## Results

We studied the effect of adding feedback connections between the first two layers in an otherwise feedforward modular network of *EI*-networks on the propagation of synchronous spiking activity along the network. Specifically, we compared the response of a purely feedforward network (FFN) with the response of a resonance pair network (RPN) to a variety of input stimulus conditions. To construct the RPN, we added feedback connections between the first two layers of the original FFN. The FFN consisted of 10 layers, each one consisting of a recurrent *EI*-network comprising 200 excitatory and 50 inhibitory neurons (Fig 1, see Methods). The input and EI-balance were adjusted such that in the baseline state, each layer of the FFN and RPN operated in an asynchronous-irregular regime in the absence of any pulse packet input and the network exhibited an asynchronous-irregular state [34, 35]. Thus, the background activity in each layer was characterized by highly irregular inter-spike intervals, low pairwise correlations, and weak network synchrony (see S1 and S2 Figs; and Methods). However, when the EI-balance was altered, either by increasing the external drive or the *EI*-ratio or by a transient input (pulse packet), each layer of the FFN or RPN exhibited damped oscillations. The natural frequency of these oscillations was determined by neuron, network and synapse parameters [40].

By the choice of parameter values in our model, the natural frequency of the damped evoked oscillations was 40 Hz.

## Pulse packet propagation in an FFN

We first tested the propagation of synchronous spiking activity by stimulating the FFN with a single pulse packet ($\alpha$ = 20 spikes, $\sigma$ = 2 ms). This mimicked earlier simulation experiments [17, 20, 36], but with different FFN parameters. Given the weak projecting synapses and sparse inter-layer connectivity, this weak pulse packet failed to propagate along the feedforward network (Fig 2a–2c). The injection of a pulse packet into the first layer network resulted in a clear but weak spike response in that layer, a much weaker response in the second layer (Fig 2b), and no tangible response in any of the subsequent layer networks. This failure to propagate was confirmed by the low signal-to-noise ratio in the 10*th* layer network (*SNR* < 4). Consistent with the weak spiking responses, there was no visible trace of the pulse packet in the subthreshold membrane potentials beyond the second layer (Fig 2c).

Next, we tested the propagation of a periodic train of pulse packets, each with the same characteristics as the single pulse packet described above. Consistent with previous results [17], such a periodic input successfully propagated along the feedforward network using the network resonance mechanism (Fig 2d and 2e, 10*th* layer *SNR* = 4.5). However, while the periodic pulse packet train did indeed successfully propagate to the last layer, this propagation was very slow. Thus, a distinct pulse packet response was observed there only after some 15 input cycles (Fig 2e), highlighting once more the key problem associated with the CTR mechanism. The reason for this is that each layer takes 2–3 cycles to build up strong enough oscillations of the membrane potentials in the next layer neurons to generate a reliable spike response.

## Pulse packet propagation in an RPN

One way to speed up activity propagation using CTR is to remove the need to build up of resonance in each layer. This can be achieved if the pulse packet can already be amplified in the second layer. To this end, we can take advantage of the network oscillations with an appropriate phase relation between the first two layers—like CTC. A simple way to induce coherent oscillations in the first two layers is to connect them in a bidirectional manner, such that they can entrain each other [26]. Therefore, we tested whether bidirectional connectivity can speed up the propagation of pulse packets.

To implement such a connectivity, we randomly selected 70 excitatory neurons from the second layer and projected them back to 70 randomly selected excitatory neurons in the first layer (Fig 2f). We made sure that the 70 neurons that projected back to the first layer were different from those that projected forward to the third layer. The synaptic strength, transmission delay, and connection probability of the feedback projections were all identical to those of the forward projections unless otherwise is mentioned in each Figure caption. We refer to the two bidirectionally connected layer networks as the *resonance pair*. Interestingly, the injection of a single pulse packet into the resonance pair network (RPN) was sufficient to initiate transient oscillations in the first and second layer networks. The bidirectional excitatory connectivity between the two layers rapidly amplified these oscillations which, once sufficiently amplified, successfully propagated to all subsequent layer networks (Fig 2g–2j, in both cases the SNR of 10*th* layer was ≈6.5).

Next, we tested whether RPN can also propagate a periodic train of pulse packets. To this end, we stimulated the first layer with a train of pulse packets (PT) and studied the interaction between the endogenous (because of the resonance pair) and the exogenous (because of the injected PT) oscillations. We found that the RPN was able to propagate the PT when the PT-

frequency was within a small range of the natural frequency of oscillations (S3a Fig). Because the transmission of the PT exploits the oscillations, propagation of a regular PT with a fixed frequency resulted in a maximum SNR. Quasi-periodic PTs (with the arrival times of pulse packets was jittering by a small amount) could also be propagated but, as expected, the SNR of such signals was weaker. Nevertheless, pulse packets jittered by 1/4 of the natural period of the network oscillations (∼6.25 ms) could still be propagated faithfully (S3b Fig).

Given the bidirectional connectivity between the first two layers, it is possible that both a single pulse packet and a regular train of pulse packets can induce sustained oscillations in the network. In our model, because we operated in the inhibition-dominated regime, recurrent inhibition prevented the emergence of sustained oscillations. Still, a single pulse packet stimulus generated oscillations that outlasted the stimulus by 8–10 oscillations cycles (Fig 3).

The number of oscillation cycles can be reduced by increasing the mean inhibition in the network. We found that by increasing the rate of Poisson input to the inhibitory neurons, the number of oscillation cycles decreased (see Figs 3, 2i and 2j). We checked that such a decrease in the number of oscillation cycles did not distort signal propagation efficiency (S4 Fig). Since increasing the Poisson input rate to inhibitory neurons caused only the number of oscillation cycles to decrease, without distorting signal propagation efficiency, we will not present the results for the case of increased Poisson input rate.

Overall, the results shown in Fig 2 demonstrate that only a small change in the network architecture, adding feedback connections between only the first two layers, can enable propagation of a single pulse packet using CTR, without driving the system into sustained oscillations. In the following, we quantify the effect of various network connectivity parameters on the network resonance and the propagation of pulse packets.

## Effect of resonance pair connectivity on pulse packet transmission

The loop transmission delay and the inter-layer connection strength are two important parameters of the resonance pair. Together, they determine whether a single pulse packet can create transient oscillations and propagate the activity along the RPN. To characterize the effect of these two parameters, we systematically varied each of them and measured the resulting *SNR* in the tenth layer of the RPN (Fig 4). First, we varied both the delay and the synaptic strength of the connections between the layers (Fig 4a). Here, we set both the delay and strength of the feedback projections to be identical to those of the feedforward projections. We found that the input pulse packets propagated most successfully when the inter-layer delay was about 12.5 ms. As the inter-layer connection strength was increased, the range of inter-layer delays for which the input pulse could propagate also increased (Fig 4a). With 12.5 ms inter-layer delay, the total loop delay for the resonance pair was 25 ms. Not surprisingly, this loop delay matched the period of the intrinsic network oscillations (corresponding to the resonance frequency of 40 Hz) of each individual layer *EI*-network.

Next, we fixed the feedforward delays at 5 ms and varied the delays of the feedback projections from layer 2 to layer 1. We found that in this case the feedback delay should be ≈20 ms to enable most successful propagation (Fig 4b). That is, most successful propagation again occurred when the loop delay (forward plus feedback delay) was 25 ms, again matching the resonance frequency (40 Hz) of the individual layer *EI*-networks.

To find the range of feedback and feedforward delays for which inputs could propagate, we varied each of these two delays independently, while keeping the inter-layer connection strength as ($J_{ee}$ = 0.33 mV, Fig 5). We found that propagation was successful for a wide range of individual feedforward and feedback delays. Once again, it was most successful if the sum of

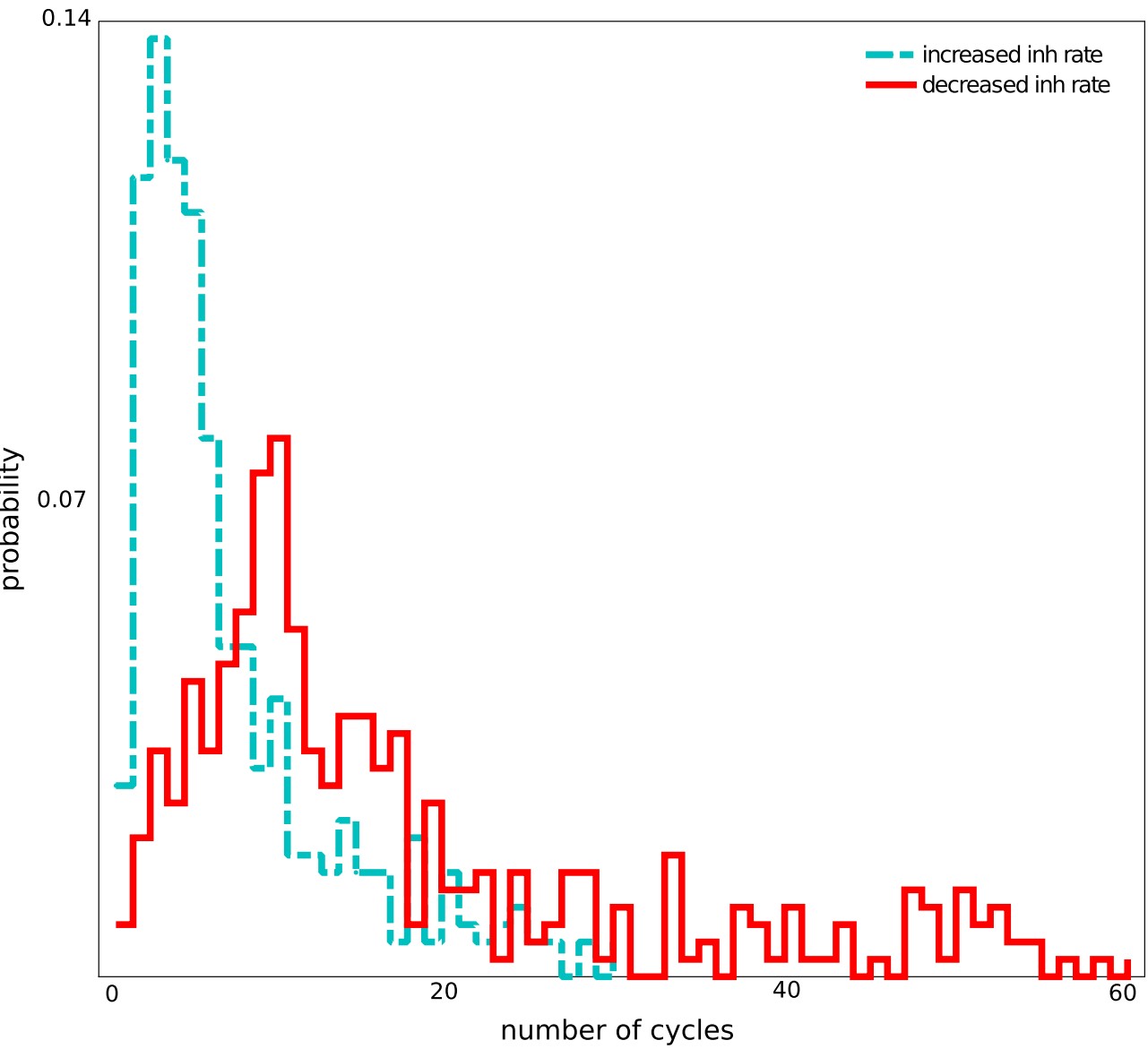

**Fig 3. Distribution of the durations of oscillatory activity in the RPN upon injection of a single pulse packet.** The RPN, when operating in a successful propagation mode, was able to quench the stimulus-induced oscillations after several oscillation cycles. The blue curve shows the distribution for an RPN with increased Poisson input rate to the inhibitory populations. Oscillation durations (shown in units of oscillation cycles) followed a distribution with *median* = 14 (for the red trace, and *median* = 5 for the turquoise trace) oscillation cycles. These data were collected from 400 trials for each simulation experiment.

the two delays (the loop delay) matched the period of the intrinsic network oscillations (here: 25 ms) of the individual layer *EI*-networks.

The above results were obtained for RPNs in which each layer was composed of 200 excitatory and 50 inhibitory neurons. To confirm that these results hold for larger networks as well, we simulated an RPN in which each layer was composed of 2, 000 excitatory and 500 inhibitory neurons. Obtaining asynchronous-irregular activity in such large RPN required fine-tuning of excitatory recurrent connection weights ($J_{ee}$ = 0.25 mV) and inter-layer connectivity (see Table 4 for the changed parameter values). With these slightly different parameters, the resonance frequency of the network was ≈33 Hz. In this large RPN, we again found stable

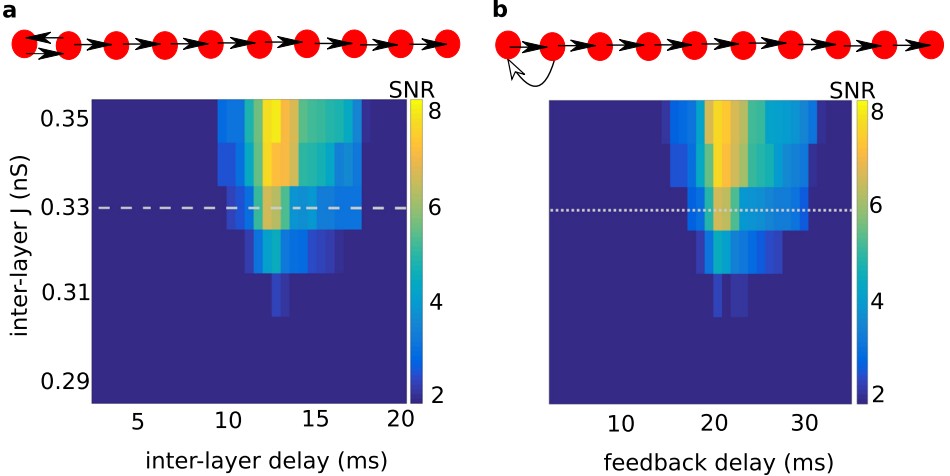

**Fig 4. Signal-to-noise ratio (*SNR*) for 10*th* layer in the RPN depends on inter-layer delays and connection strengths of the resonance pair.** **(a)** Delays for feedforward and feedback connections were set equal to each other and were systematically varied along the X-axis. Note that the most successful propagation was observed for a total loop delay (forward plus feedback delay) of 25 ms, matching the period of the intrinsic resonance oscillation of each individual layer *EI*-network (resonance frequency of 40 Hz). The range of inter-layer delays for which propagation was successful expanded as the inter-layer connections were strengthened. However, the *SNR* was still considerable for weaker ones. **(b)** Delays for feedforward connections were fixed to 5 ms, and for feedback connections were systematically varied along the X-axis. Again, the most successful propagation was observed for a total loop delay of 25 ms, matching each individual layer *EI*-network's resonance frequency of 40 Hz. In the schematic representations of the network structure (top panels), the length of the arrows indicate the duration of inter-layer delays. The dashed and dotted horizontal lines in **(a)** and **(b)** indicate the value of $J_{ee}$ used to represent successful propagations in other figures. A similar plot has been provided for the case of increased Poisson input rate to the inhibitory subpopulations in S4 Fig.

propagation of pulse packet, assisted by the resonance pair (S5a Fig). Next, we checked how the transmission was affected by the recurrent excitatory connection strengths ($J_{ee}$) and inter-layer connection delays (S5b Fig). Consistent with the results obtained for the smaller network (Fig 4a), in the large network the resonance and strongest transmission was obtained for an inter-layer delay of ≈15 ms, once again matching the network resonance frequency (S5b Fig). These results confirm that successful signal propagation primarily depends on the resonance pair's loop delay, which should be consistent with the network resonance frequency.

Based on these results (Figs 4 and 5), we conclude that inter-layer connection delays should match the resonance frequency of the resonance pair networks, but how precisely these delays should be tuned is another issue. To study this, we simulated the RPN in which both within- and inter-layer delays were chosen from a Gaussian distribution whose mean was set to 1.5 ms (for within-layer connections) and 12.5 ms (for inter-layer connections). The standard deviation of the delay distribution was set to either 10% or 20% of their respective mean values. We found that pulse packets propagated successfully when the standard deviation of the delays was 10% of the mean value, but failed to propagate for larger standard deviations (S6 Fig).

## Resonance pair improved both the threshold and speed of propagation of pulse packets

Next, we addressed the question to what extent the inclusion of feedback *EE* connections between the first two layer networks of the FFN affects the threshold and speed of propagation of pulse packets in the network. To this end, we compared both the speed and *SNR* of the pulse packet response in the FFN and the RPN. For this comparison, we stimulated the RPN with a single pulse packet, whereas the FFN was stimulated with a periodic train of pulse packets

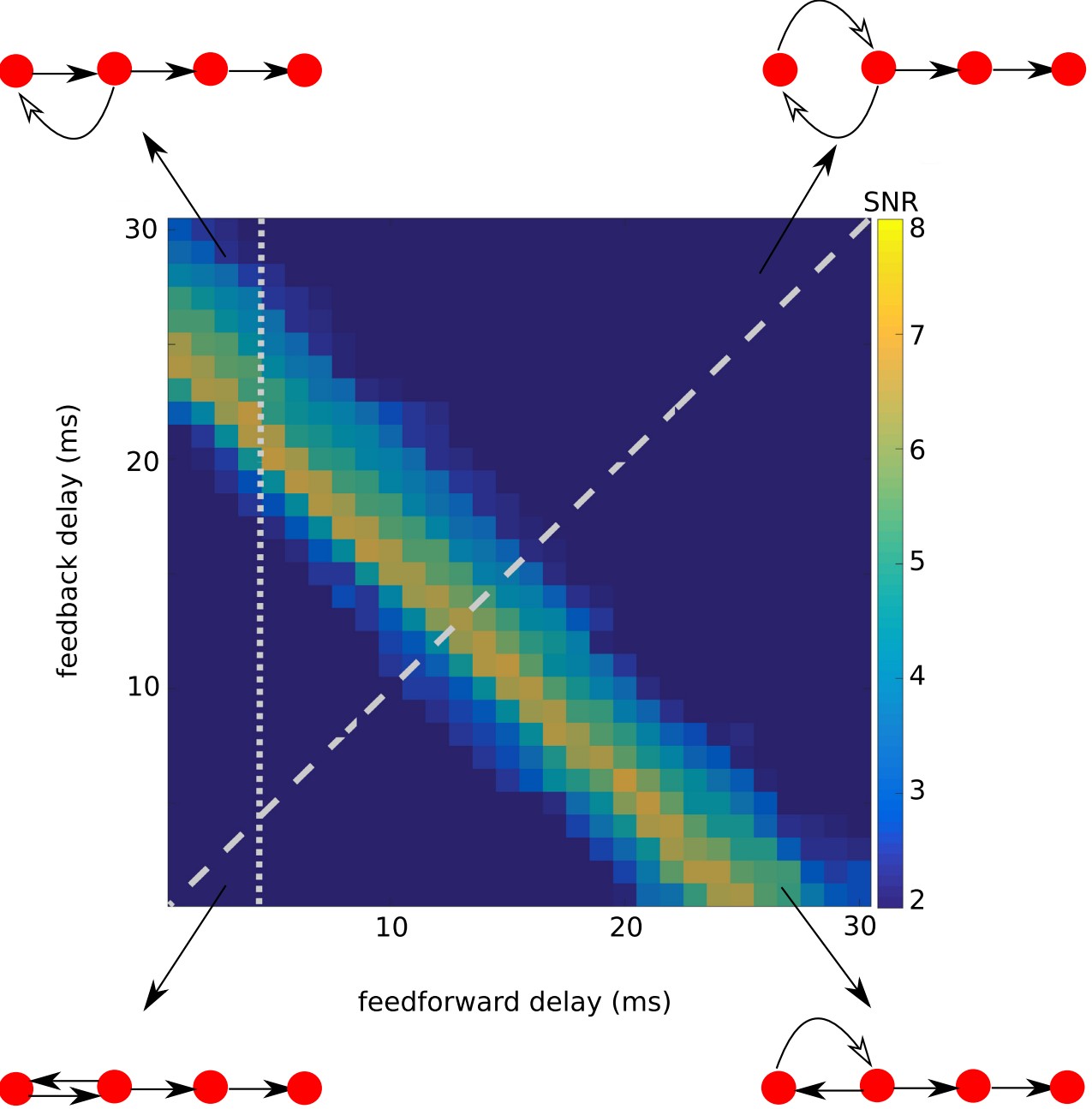

**Fig 5. Signal-to-noise ratio (*SNR*) for 10*th* layer in the RPN for independently varied feedforward and feedback delays.** The sum of the feedforward and feedback delays is the key parameter to enable signal propagation. When the inter-layer connection strength, $J_{ee}$, was fixed at 0.33 nS, most successful propagation was obtained for the condition that the sum of forward and feedback delays, rather than any of their individual values, matched the resonance period of the individual layer *EI*-network's resonance frequency of 40 Hz. In the schematic representations of the networks, only the first four layers are depicted, with the length of the arrows representing the delays between the resonance pair layer networks.

(Fig 6a and 6b). The loop delay of the resonance pair in the RPN and the inter-pulse intervals in the periodic stimulation of the FFN were matched the resonance period of the *EI*-networks in the resonance pair, layers 1 and 2. We found that introducing feedback projections substantially increased the *SNR* of the pulse packet response in the RPN as compared to that in the

**Table 4. Synapse and connectivity parameters for the large network.**

| Name | Value | Description |
|---|---|---|
| $\tau_{exc}$ | 1 ms | Rise time of excitatory synaptic conductance |
| $\tau_{inh}$ | 1 ms | Rise time of inhibitory synaptic conductance |
| $E^{exc}_{syn}$ | 0 mV | Reversal potential of excitatory synapses |
| $E^{inh}_{syn}$ | -80 mV | Reversal potential of inhibitory synapses |
| $J_{ee}$ | 0.25 mV | Exc. to exc. synaptic strength measured at -70 mV |
| $J_{ei}$ | 1.5 mV | Exc. to inh. synaptic strength measured at -70 mV |
| $J_{ie}$ | -6.2 mV | Inh. to exc. synaptic strength measured at -54 mV |
| $J_{ii}$ | -12.0 mV | Inh. to inh. synaptic strength measured at -54 mV |
| $J_{pe}$ | 0.25 mV | Connection strength: Input Poisson spike train to exc. pop. |
| $J_{pi}$ | 0.4 mV | Connection strength: input Poisson spike train to inh. pop. |
| $J_{pp}$ | 0.25 mV | Connection strength: Pulse packet to P neurons in first layer |
| $d_{within-layer}$ | 1.5 ms | Transmission delay within layer |
| $d_{inter-layer}$ | 30—33 ms | Range of total resonance delay between layers |
| $N_{exc}$ | 2000 | Size of excitatory population per layer network |
| $N_{inh}$ | 500 | Size of inhibitory population per layer network |
| $N_{proj}$ | 680 | Number of projecting neurons per layer network |
| $\epsilon_{within-layer}$ | 0.2 | Connection probability within-layer network |
| $\epsilon_{inter-layer}$ | 0.1 | Connection probability between layer networks |

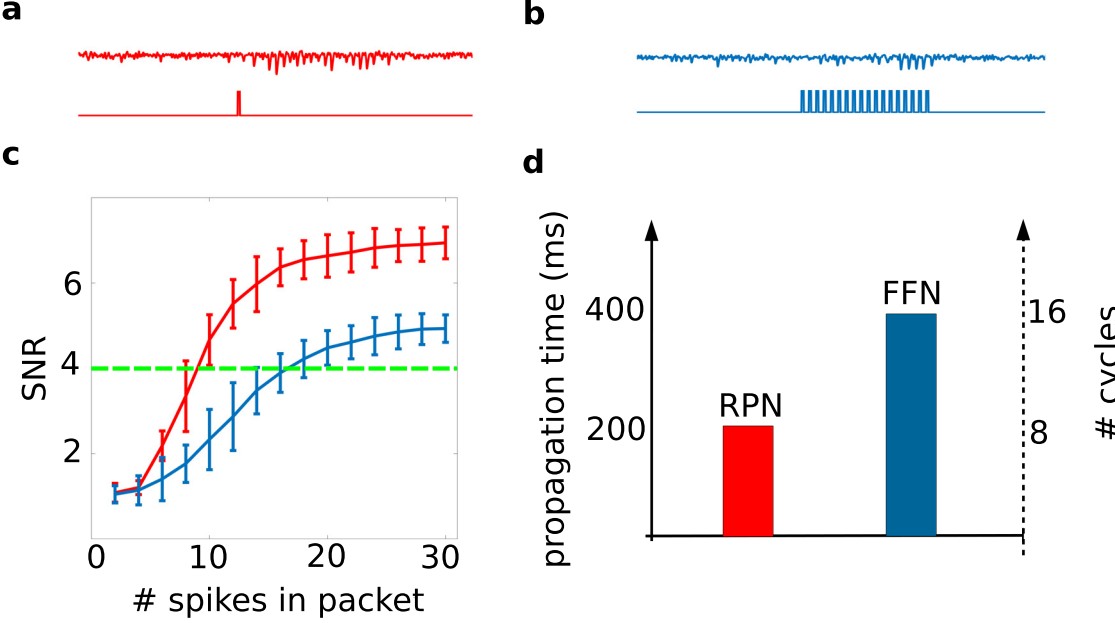

**Fig 6. Introducing a resonance pair improves both the threshold and speed of propagation of synchronous spiking. (a)** Averaged membrane potential of *E* neurons in the 10*th* layer in response to a single pulse packet (depicted in bottom trace) in the RPN, in the presence of feedback projections from layer 2 to layer 1. **(b)** Averaged membrane potential of *E* neurons in the 10*th* layer in response to a periodic pulse packet (depicted in bottom trace) in the FFN, in the absence of feedback projections. **(c)** *SNR* in the 10*th* layer of the RPN (red curve) and FFN (blue curve) as a function of strength, *α*, of the input pulse packet. Increasing *α* increased the *SNR* for both RPN and FFN. However, the red curve crosses the green dashed line (as an arbitrary threshold for successful propagation) at a clearly smaller value of *α* than the blue curve, implying clearly lower threshold of synchrony propagation in the RPN. **(d)** On average, synchronous activity propagates much faster in the RPN, by at least a factor of two, than in the FFN.

FFN (Fig 6c). This meant that much weaker pulse packets could propagate in the RPN than in the FFN. Thus, adding sparse *EE* feedback connections between only the first two layers of the FFN significantly reduced the threshold (minimum value of pulse packet strength $\alpha$) for successful propagation throughout the entire FFN.

Next, we compared the propagation velocities in the RPN and the FFN. For a fair comparison of propagation speed in these two networks, we set the forward transmission delays to 5 ms in both networks. Therefore, to meet the condition that the loop delay in the resonance pair should match the intrinsic resonance in the participating *EI*-networks, the feedback delay was set to 20 ms in the RPN. In the FFN, as noted before, the pulse packet needed to be recreated by gradual build-up in each layer successively. Hence, it took on average between 2–4 oscillation cycles in each layer, before the pulse packet successfully reached the next layer. As shown in Fig 6d, the bidirectional projections between the first two layers in the RPN sufficed to rapidly amplify the network response, and, hence, there was no need to gradually build-up and recreate the pulse packet in each individual layer. As a result, the transmission in the RPN was much faster, by at least a factor of two, than in the FFN. These results demonstrate that introducing sparse feedback projections from layer 2 to layer 1 in an FFN with weak and sparse connections substantially accelerates the propagation of synchronous spiking in such network, thereby alleviating a significant problem associated with the mechanism of *communication through resonance*.

## Network background activity

For stable propagation of synchronous spiking activity, it is important that the ongoing activity of the network remains stable and exhibits an asynchronous-irregular state without population activity oscillations [36]. In principle, the feedback projections in the resonance pair could destabilize the asynchronous-irregular activity state, induce spontaneous oscillations, and lead to the propagation of random fluctuations in the network activity. Therefore, we measured the effect of the feedback and feedforward projections on the network background activity. The strengths of feedforward and feedback connections in the RPN were set to be identical. First, we systematically varied the inter-layer connection strength and the rate of external (excitatory) input, and measured the population activity synchrony (population Fano factor, *pFF*) for the 10*th* layer of both the RPN and the FFN (Fig 7a and 7b). We also compared the firing rates, the irregularity of spike timing (*CV*) and the pairwise correlations for three different choices of these two parameters (S2 Fig).

We found that for weak external inputs, the background network activity remained in an asynchronous-irregular regime in both the RPN and FFN for a wide range of inter-layer connection strengths (Fig 7a and 7b). Likewise, for weaker inter-layer connections, the background network activity of both the RPN and FFN remained in an asynchronous-irregular regime. However, when both external input and inter-layer connections were strong, large fluctuations induced by the external input could propagate to downstream layers. Propagation of such spurious fluctuations resulted in synchronous-irregular activity in the downstream networks (Fig 7a and 7b, and S2 Fig; see also S1 Fig for raster plots). Such undesirable emergence of synchrony in the background network activity because of stronger inter-layer connections and stronger external input was observed in both the RPN and FFN. However, in the RPN this transition to synchronous-irregular background activity occurred at clearly lower values of external inputs and inter-layer connection strengths than in the FFN (compare Fig 7a and 7b). That is, while the resonance pair reduced the threshold for propagation and accelerated the pulse packet propagation, it also constrained the range of network and input parameters for which stable propagation could be observed.

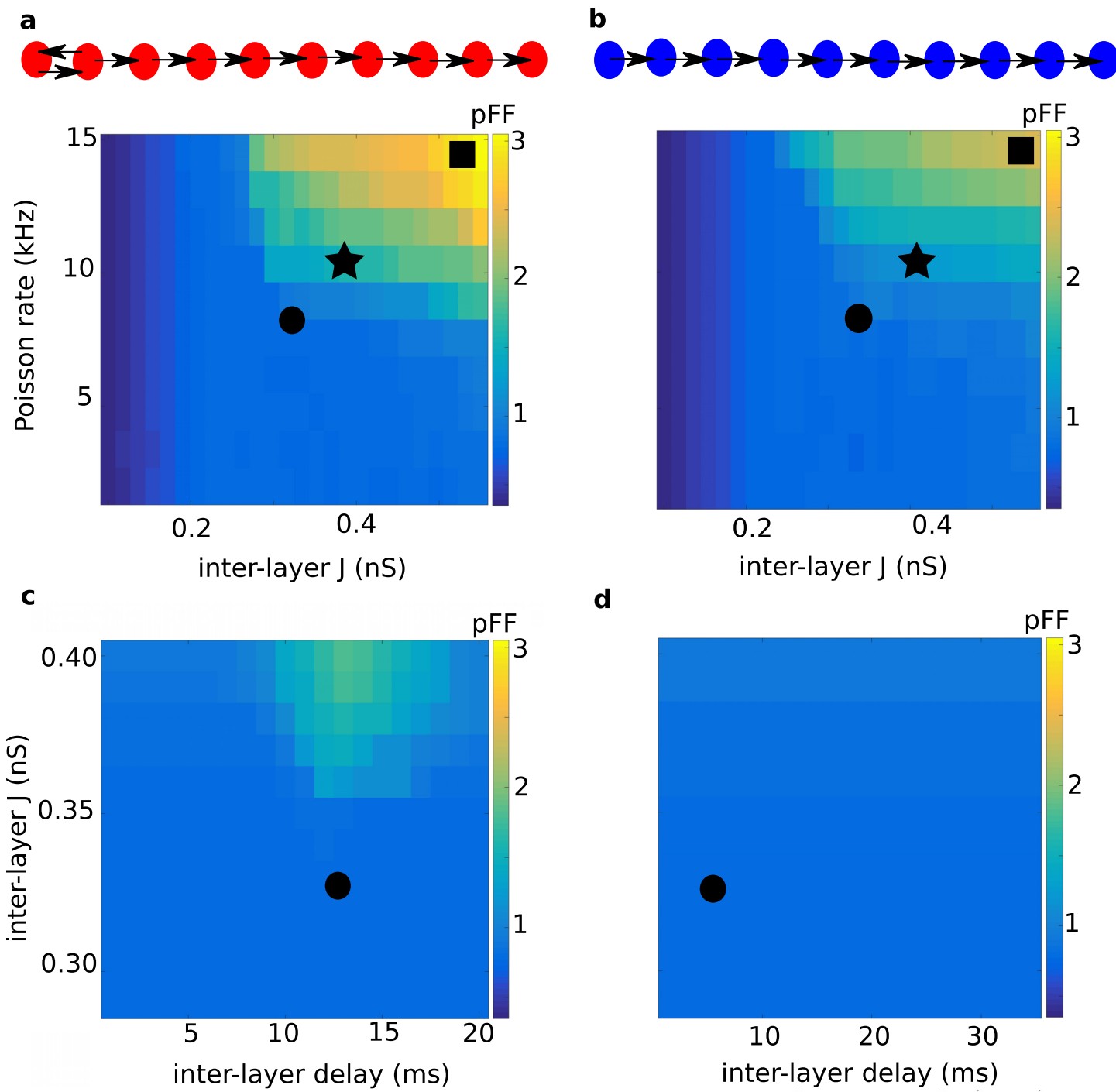

**Fig 7. Different background activity regimes in the RPN (a, c) and FFN (b, d) networks.** The population Fano factor in the 10*th* layer of the RPN **(a)** and FFN **(b)**, is shown as a measure of synchrony in the background network activity for different strengths of inter-layer connections (X-axis) and input rate (Y-axis). The cyan area, indicated by an asterisk, denotes a synchronous irregular regime, whereas the vast, blue area denotes the asynchronous irregular regime, with a long-tailed distribution of $CV_{ISI}$ and low average correlation coefficients (S2 Fig). Both network types transit to the synchronous irregular regime, indicated by a black square, with increasing input rate and inter-layer connection strength. However, the RPN reaches the synchronous irregular state much earlier than the FFN. The population Fano factor in the 10*th* layer of the RPN **(c)** and FFN **(d)**, is shown for different inter-layer connection delay (X-axis) and strength (Y-axis). The input rate was set to 8 kHz for both network types. For strong enough inter-layer connections, provided their loop delay matched the resonance period of the network, sustained background activity oscillations might develop in the network and propagate to the downstream layers. Black circles in all four panels indicate the parameter settings used to investigate the pulse packet propagation in Figs 2 and 6 and the red trace in Fig 3. In panels **(a)** and **(b)**, the feedforward and feedback delays were set to 5 ms, respectively.

To determine the degree of synchrony in the background network activity for different inter-layer connection strengths and delays, we measured the population Fano factor (cf. Methods) for both the RPN and FFN networks, with the input rate set to 8 kHz (Fig 7c and 7d). These results demonstrate that the inter-layer delay plays no role in inducing synchrony in the FFN background network activity (Fig 7d). However, it does render a regime for eliciting synchronous background activity in an RPN (Fig 7c). This regime existed for the range of delays that matched the resonance period of the *EI*-networks involved, and for stronger inter-layer connections it increased significantly. Therefore, the parameter values causing this synchronous regime in the RPN background activity should be carefully avoided, because this regime prohibits reliable signal propagation.

## Conditions for resonance in the resonance pair

The connectivity between the layers of the resonance pair could affect the propagation of synchronous spiking in the RPN in different ways. It could prohibit the propagation of pulse packets by enabling spurious network fluctuations to propagate, or by altering the resonance properties of the two layer networks involved. Whereas weak connectivity may not allow the resonance to occur, strong connectivity could induce spontaneous network oscillations, precluding the resonance-based mechanism from supporting the propagation of pulse packets. Therefore, we systematically varied both the forward and feedback connectivity between the two layers and determined the regime most suitable for communication through resonance (Fig 8). We found that an increase in either the connection probabilities (Fig 8a and 8b) or connection strengths (Fig 8c and 8d) increased the network's propensity to oscillate. Strong feedback connections and high connection probabilities induced spontaneous oscillations in both layer networks. The diagonal symmetry of Fig 8a and 8b (and to a lesser extent in Fig 8c and 8d) shows that the feedback connections can compensate for a lack of feedforward connections (as in Fig 8a and 8b), or their weakness (as in Fig 8c and 8d). For moderate values of the feedback connection probability and connection strength, there is a region in the parameter space for which single pulse packets can be propagated by exploiting the network resonance property, without destabilizing the network activity dynamics into sustained network oscillations. This region is distinguished by a *pFF* of about 1, the blue area in Fig 8a and 8b, and an example of it is marked with a black circle in all four panels of Fig 8.

## Discussion

Oscillations are an ubiquitous feature of the activity of neuronal populations and are assumed to serve many functions. An important function attributed to $\alpha$ (8–12 Hz), $\beta$ (12–30 Hz) and $\gamma$ (30–80 Hz) oscillations is that they help in communicating spiking activity between weakly connected networks [9, 12]. Oscillations in different bands can be combined to form various strategies for flexible gating of activity [41]. *Communication through coherence: CTC* [12] and *communication through resonance: CTR* [17] are two related mechanisms by which oscillations can influence communication between neuronal networks. Oscillations in both mechanisms, modulate the excitability of neurons in the population receiving the signal—the spiking activity of the sender population. Signals which impact a population at the right time within its high excitability period can affect the spiking activity of the receiver neurons and, thereby, increase their chance to be propagated. Unlike CTC, the CTR mechanism is based on evoked oscillations and does not require spontaneous coherent oscillations between sender and receiver networks. However, CTR based communication is slow because it is based on the gradual build-up of the evoked activity over several oscillation cycles in the receiver population. Moreover, only trains of pulse packets (either periodic of the right

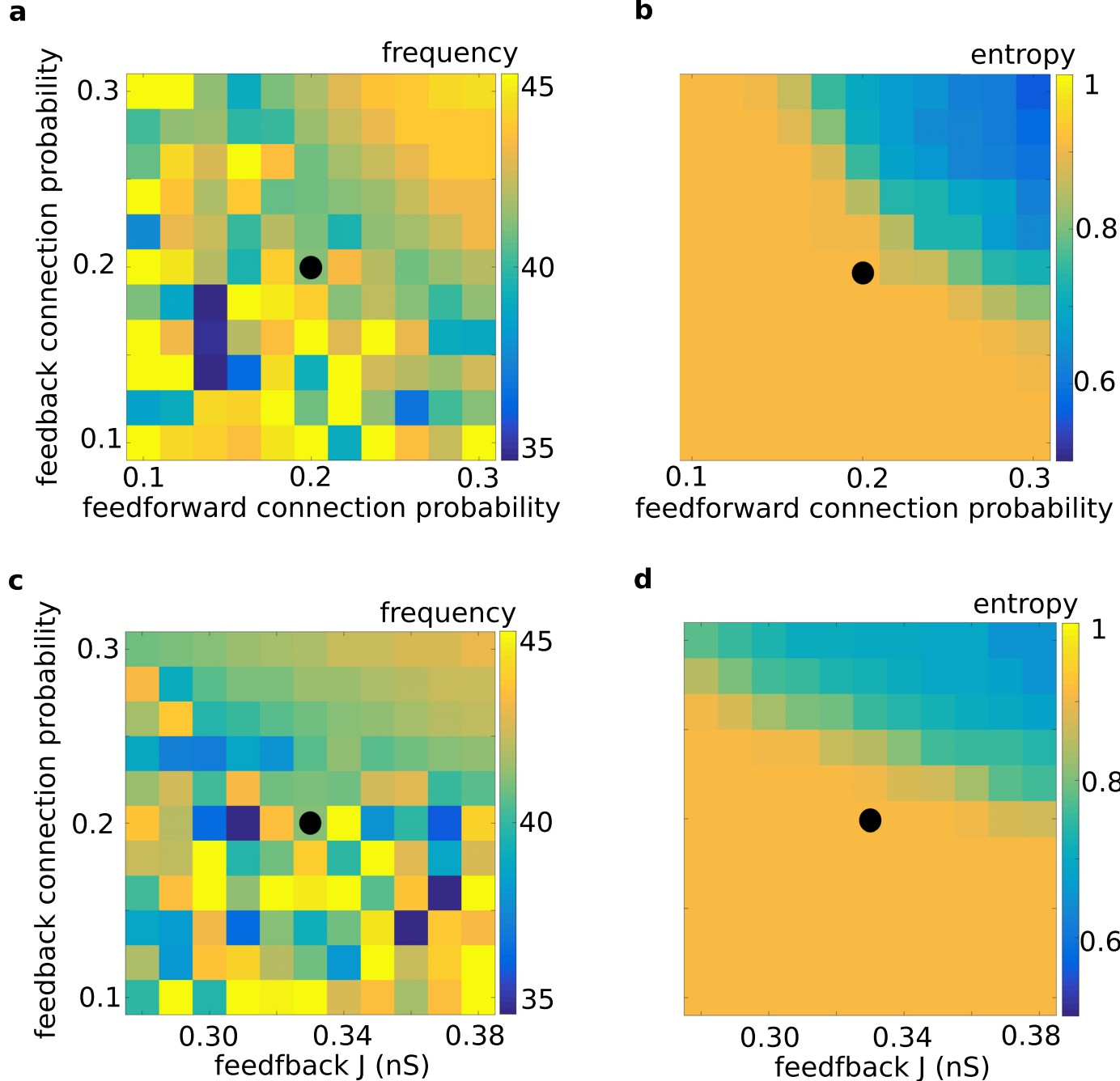

**Fig 8. Conditions for resonance in a bidirectionally connected two-layer network.** Network resonance frequency and spectral entropy were calculated as a function of the feedback and feedforward connection probability (**a, b**), and as a function of the feedback connection probability and strength (**c, d**). Both an increase in the feedback connection probability and strength increased the propensity of the network to exhibit resonance. However, when the feedback connections were too numerous or too strong, the network exhibited sustained oscillations as the network dynamics bifurcated to the synchronous irregular state. This state, represented by lower values of spectral entropy in (**b**) and (**d**), started with a population oscillation frequency of around 40 Hz, which gradually increased to 43 Hz (**a, c**). Note that at higher values of spectral entropy, the frequency of the oscillations was not well-defined and did not have a consistent value: The oscillation frequency appeared noisily in the regime where oscillations were weak (high spectral entropy; see panels **b** and d) and, therefore, it is difficult to determine the peak frequency. Once the network entered an oscillatory regime (low spectral entropy), the peak frequency estimate became more reliable. Black circles in all four panels indicate the parameter set used in Figs 2 and 6 and the red trace in Fig 3 for investigating the pulse packet propagation in the absence of sustained oscillations.

frequency or aperiodic of high enough rate) can be transmitted by this mechanism, but the propagation of a single pulse packet is not feasible [17]. Here, we addressed these problems of CTR and showed that introducing bidirectional connections between two upstream modules in an otherwise feedforward network can enable the propagation of single pulse packets and can also significantly speed up the propagation. This increase in propagation speed was achieved because the two bidirectionally connected layers amplified the pulse packet to a level at which they could be propagated through the successive layers without the need for further amplification.

Reverberation of the transient stimulus between the first two layers of the network, the resonance pair, fed the downstream remainder of the network with a temporally coordinated and strong train of pulse packets, with inter-packet intervals determined by the sum of the forward and backward transmission delays of the resonance pair. Hence, matching the two internal time constants of the system, the resonance period of the individual modules and the loop transmission delay of the resonance pair, sufficed to enable the reliable propagation of a single pulse packet across the entire network through the built-in CTR mechanism. Indeed, in a series of simulation experiments we could demonstrate that in our network model, the consistency of the two time scales, determined by the intra-module lateral (recurrent) connections, and inter-module feedforward and feedback connections, facilitates the transmission of transient synchronous spiking signals.

## Bidirectional connectivity between networks in the brain

Based on the available data on anatomical and functional connectivity in the brain, it is not easy to determine whether connections between networks (at mesoscopic scale) in the brain are unidirectional or bidirectional. Mesoscopic anatomical connectivity measured by DTI [31] or by tracer injections [2] suggests that connectivity among most pairs of networks is bidirectional. But this does not necessarily mean that these networks are effectively bidirectionally connected. This is because of a number of reasons: 1. It is not clear whether connections are equally strong in both directions—DTI and tracer injection techniques are not well suited to determine this. In fact, data from trace studies when thresholded suggest that connections are clearly weaker in one direction than in the other ([2, 42–44]). 2. Selective stimulation of a given brain region does not seem to evoke reverberating activity, as would be expected from bidirectional connections [42–44]. 3. Historically also, starting from the description of the visual information processing areas [28] to the latest mesoscopic connectivity studies of the mouse brain [2, 31, 45, 46], there is a widespread consensus of a hierarchical arrangement of brain network connectivity. Thus, it seems reasonable to assume that most connections are effectively unidirectional and only few pairs of networks are effectively connected in a truly bidirectional manner.

For the aforementioned reasons, in this study we restricted our investigation to a case in which only a single pair of networks were connected in a bidirectional manner. However, it is worth asking what would happen if more, or even all, networks in an FFN were connected in a bidirectional manner. Based on the results shown in Figs 7 and 8 we expect that, unless feedback and feedforward connections (both delays and strengths) are carefully adjusted, all layers will show sustained oscillations. Indeed, our network simulations confirmed this. Such a sustained oscillatory state is neither biologically plausible nor is it suitable for activity propagation. While it may be possible to find connectivity parameters that provide a near asynchronous-irregular activity state in a fully bidirectionally coupled network, such an investigation clearly demands a separate, more dedicated model study.

### A functional role of feedback projections

Feedback projections play a role in regulating neuronal network activity [47, 48], brain activity oscillations [49–51], and high level brain functions such as working memory [52], vision [53, 54], attention [55, 56], and consciousness [57–59]. Here, we studied how feedback connections can help improve the propagation of synchronous spiking activity in feedforward neuronal networks. We showed that including a pair of bidirectionally connected modules into an otherwise feedforward network promotes the propagation of synchronous spike volleys in the network.

The possible role of feedback connections in the propagation of synchronous pulse packets through modular networks has been suggested earlier by Moldakarimov et al. [60]. There, it was shown that feedback connections increased the number of spikes in the synchronous spike volley and, thereby, helped the pulse packet propagate in the feedforward network [60]. That mechanism, however, operates on a much shorter time scale than the one we propose here. In their model [60], propagation was facilitated by feedback delays within the temporal spread of the injected pulse packet, i.e., up to only few milliseconds. The mechanism we propose here is both qualitatively and quantitatively different and is based on the resonance property of the *EI*-networks involved in the feedforward network. Here, the impact of a pulse packet on the target *EI*-network provides, thanks to the damped resonance oscillation it evokes, a short range of specific time windows with enhanced excitability and, hence, larger response to the next incoming pulse packet. As a result, the reverberation of the pulse packet between the bidirectionally connected layer networks in the resonance pair builds up even stronger pulse packets for the downstream layers of the network. We found that a prerequisite for successful propagation of such synchronous spiking activity was that the loop transmission delay in the resonance pair (forward plus feedback delay) matched the resonance period of the individual layer EI-networks.

### Possible applications in bottom-up and top-down information transfer

Recent studies have suggested different functional roles of high and low frequency oscillations in bottom-up and top-down signaling in cortical networks [11, 61]. It has been shown that the transmission of information along the feedforward pathway from peripheral sensory areas to higher areas in the cortical hierarchy is mainly carried by gamma range oscillations, whereas feedback signals are mostly conveyed by alpha and/or beta oscillations [11, 62–64]. These results gained support from experimental observations of strongest synchronization in the gamma band in superficial cortical layers, whereas synchronization in the alpha-beta band was found to be strongest in infragranular layers [65]. In our network model, the baseline activity of the layer networks lacked spontaneous oscillations, but they exhibited a resonance property in the low-gamma range. The presence of a single feedback loop with matching loop delay resulted in short-lived gamma oscillations upon transient stimulation of the first layer network, resulting in reliable signal propagation throughout the entire feedforward pathway, consistent with the above-mentioned experimental observations for bottom-up transmission. Incorporating further feedback loops between the downstream layer networks with different resonance frequency, possibly in beta range, can provide a more complete model for explaining forward and backward signaling in cortical networks of networks.

### Relationship between CTR and CTC

To facilitate transmission of spiking activity between two weakly connected networks, CTC suggests that coherent oscillations between two networks can periodically modulate the effective coupling between networks and a suitable phase relation between the spontaneous

oscillations of the two networks can facilitate the exchange of signals between them [12]. In CTR we assume that oscillations are not spontaneously generated but that they are evoked by the incoming pulse packets themselves. Evoked oscillations are then amplified by successive incoming pulse packets, exploiting the resonance property of the receiving network. If their timing matches the network resonance frequency, and once the oscillations are strong enough, the activity propagates to the receiver network. Thus, in essence, CTR and CTC are quite similar: Both are based on changes in the excitability of the receiver network and both require a suitable phase relation between the oscillations of sender and receiver networks. The crucial difference between the two is that CTC requires spontaneous coherent oscillations between sender and receiver networks, whereas in CTR, oscillations are stimulus-evoked and, hence, emerge only upon arrival of the incoming stimulus. Moreover, unlike in CTC, in CTR the suitable phase relation for transmission is naturally established by the emergence of oscillations because the sender activity evokes the oscillations in the receiver network. By contrast, in CTC, the mechanism underlying the coherence between sender and receiver networks, especially when oscillation frequency and phase can fluctuate, are still not well understood.

Recently, Palmigiano et al. [26] showed that two weakly connected networks of spiking neurons can show coherent spontaneous transient oscillations. Such oscillations can form the basis for CTC, provided the sender and receiver networks are tuned to show spontaneous oscillations and the networks operate around the border between non-oscillatory and oscillatory activity regimes. In such networks, when transient oscillations spontaneously appear, the weak connections ensure that the two networks synchronize. By contrast, in RPN the network parameters are set such that every layer operates in an asynchronous regime and does not show any spontaneous oscillations (S2 Fig). Instead, oscillations in RPN are initiated by the incoming pulse packets and maintained for only a few cycles by the reverberation of activity between the sender and receiver networks. Such reverberations occur because of bidirectional connections, the loop-delay of which is near to the period of the intrinsic network oscillations. Thus, there are clear differences in the way oscillations are synchronised in the model proposed by Palmigiano and colleagues [26] and our resonance pair model.

Finally, CTC requires coherent oscillations between all successive layer networks. By contrast, in the RPN, activity is already amplified in the first two layer networks and no synchronous oscillations are needed to transmit the activity from the second layer network onwards. Thus, despite the apparent similarity between both mechanisms (the need for network oscillations), there are several crucial differences between CTC and CTR.

It is not straight-forward to determine whether the brain uses CTC or CTR. The ability of cortical networks to show coherent oscillations makes a compelling case for CTC. In a similar vein, though, cortical networks do show resonance properties [66]. That is, cortical networks have the necessary neuronal hardware to generate resonance properties, necessary for CTR. Possibly, the existence of coherent oscillations before the onset of a stimulus (to be transmitted) and a tight relationship between spike timing and oscillation phase would be a clear evidence for CTC [67, 68]. However, there is also experimental evidence suggesting that oscillations are not immediately visible at stimulus onset [69–71], consistent with the CTR hypothesis. We conclude that possibly both modes of network communication are being used, depending on brain areas involved and on task and behavioral context.

## Supporting information

**S1 Fig. Raster plots for three different background firing regimes of the RPN and FFN.**
Increasing input rate and inter-layer connection strength both increased the propensity of the RPN and the FFN to synchronize their background activities. For the regime marked with the

black square (rightmost column), both networks showed network activity oscillations.
(EPS)

**S2 Fig. Distributions of $CV_{ISI}$, pair-wise correlations, and firing rates of excitatory neurons in three different background firing regimes of the RPN and FFN.** Distributions of CV of inter-spike intervals (left), pairwise correlations (middle), and firing rates (right) for three different sets of external input and inter-layer connection strengths. Red and blue traces denote RPN and FFN network structures, respectively. Three states are introduced in Fig 7 with corresponding markers. For weak external inputs and inter-layer connection strengths, the network in both structures exhibited asynchronous irregular activity. In this state, adding excitatory feedback connections did not affect the network activity states. However, when the network was operating in a synchronous irregular activity state (corresponding to the higher external excitatory input and stronger inter-layer synapses, bottom row, indicated with a black square) adding feedback connections resulted in increased firing rates and synchrony indices, even more so in the RPN than in the FFN (compare red and blue traces in the two right-most panels in the bottom row), while spiking became distinctly more regular in both network types (left panel).
(EPS)

**S3 Fig. Filtering property of the RPN when injected with a periodic PT with matching intervals (a), and its robustness against deviations from periodicity (b). (a)** In the presence of both the endogenous (due to the RP) and the exogenous (due to the PT) resonance properties, the network could propagate the signals with matching inter-pulse intervals. Inter-layer delays were chosen to match the resonance period of the network. The green dashed line represents the lowest amount of SNR (= 4) above which propagation can be considered as successful. **(b)** The RPN was excited by PTs with some degree of deviation from periodicity (jitter, represented in the X-axis). For large values of jitter, the RPN failed to propagate the input PT, because the injected PT and its reverberations laid in the less responsive window of the network, while the PTs with small jitters propagated.
(EPS)

**S4 Fig. Dependence of signal-to-noise ratio (*SNR*) of 10*th* layer in the RPN with increased Poisson input rate to inhibitory populations on inter-layer delays and connection strengths of the resonance pair. (a)** Delays for feedforward and feedback connections were set equal to each other and were systematically varied along the X-axis. Like Fig 4, the most successful propagation occurred when the total loop delay (forward plus feedback delay) was 25 ms, matching the period of the intrinsic resonance oscillation of each individual layer *EI*-network (resonance frequency of 40 Hz). **(b)** Delays for feedforward connections were fixed to 5 ms, and for feedback connections were systematically varied along the X-axis. Again, the most successful propagation was observed for a total loop delay of 25 ms, matching each individual layer *EI*-network's resonance frequency of 40 Hz. This plot is the counterpart of Fig 4 in the main text, but for an increased Poisson input rate to the inhibitory neurons in the RPN. These plots emphasize that increasing the input rate to the inhibitory population does not impair signal propagation, and hence the SNR of the RPN.
(EPS)

**S5 Fig. Propagation of a PP across in a large RPN (a), and SNR of a large RPN when inter-layer connection strengh and delay are changed (a).** Here, we simulated an RPN with 2, 000 excitatory and 500 inhibitory neurons in each layer. For details of parameters see Table 4. **(a)** The first layer of a large RPN was stimulated with a PP and it propagated to the 10*th* layer of the network within 2–3 oscillation cycles. **(b)** SNR of the 10*th* layer of the large-scale RPN as a

function of the inter-layer delay and inter-layer excitatory connection strength. For an inter-layer delay of $\approx$15 ms, the SNR reached its maximum. The delay range for which successful signal transmission was observed increased by strengthening the inter-layer connections. (EPS)

**S6 Fig. Distributed delays may distort PP propagation in the RPN.** Depending on the degree of dispersion of delays (inter-layer and within-layer), PP propagation may be impaired. For these simulation examples, within-layer and inter-layer delays for each connection was chosen from a Gaussian distribution. The mean of the Gaussian distribution was set to 1.5 ms for within-layer delays and 12.5 ms for inter-layer delays. **(a)** Propagation of a pulse packet was successful when the standard deviation of the delays distribution was set to 10% of the mean, i.e., for inter-layer delays *std.* = 1.25 ms and for within-layer delays *std.* = 0.15 ms. **(b)** Propagation of a pulse packet failed when the standard deviation of the delays distribution was set to 20% of the mean, i.e., inter-layer delays *std.* = 2.5 ms and within-delay *std.* = 0.30 ms. (EPS)

## Acknowledgments

We thank Uwe Grauer and Bernd Wiebelt for helping making these HPC-facilities available to us.

## Author Contributions

**Conceptualization:** Ad Aertsen, Arvind Kumar, Alireza Valizadeh.

**Formal analysis:** Hedyeh Rezaei.

**Funding acquisition:** Ad Aertsen, Arvind Kumar, Alireza Valizadeh.

**Investigation:** Hedyeh Rezaei.

**Methodology:** Hedyeh Rezaei, Arvind Kumar.

**Project administration:** Alireza Valizadeh.

**Supervision:** Ad Aertsen, Arvind Kumar, Alireza Valizadeh.

**Visualization:** Hedyeh Rezaei.

**Writing – original draft:** Arvind Kumar.

**Writing – review & editing:** Hedyeh Rezaei, Ad Aertsen, Arvind Kumar, Alireza Valizadeh.

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
