## [Decision Letter · Decision Letter 0]

17 Oct 2019

Dear Dr Kumar,

Thank you very much for submitting your manuscript 'Facilitating the propagation of spiking activity in feedforward networks by including feedback' for review by PLOS Computational Biology. Your manuscript has been fully evaluated by the PLOS Computational Biology editorial team and in this case also by independent peer reviewers. The reviewers appreciated the attention to an important problem, but raised some substantial concerns about the manuscript as it currently stands. While your manuscript cannot be accepted in its present form, we are willing to consider a revised version in which the issues raised by the reviewers have been adequately addressed. We cannot, of course, promise publication at that time.

Sincerely,

Samuel J. Gershman

Deputy Editor

PLOS Computational Biology

[LINK]

Reviewer's Responses to Questions

**Comments to the Authors:**

Reviewer #1: The authors further explore the Communication-through-Resonance mechanism for feed-forward propagation of input pulses through a chain of network modules, by exploring solutions to make transmission faster and more efficient, by being enabled by a smaller number of input pulses. To do so, they modify the original fully feed-forward network architecture, by introducing a "resonance loop" at the initial stage of the chain, which boosts the ignition of the propagation mechanism by facilitating resonance. I personally found the manuscript very interesting and also quite well written with a very pedagogic and natural sequence of presentation of the different results. Also comparisons with the original feed-forward architecture are systematically performed, making transparent the presentation of the effects of the novel ingredient. I am therefore positive about the possibility for this study to be published after revision.

I have however some questions/suggestions for discussion/improvement.

MAYOR ISSUES:

1a) A first question is about Figure 3 shows a distribution of the number of oscillatory cycles induced by the presentation of a single pulse at the input stage. It peaks around 10-12 cycles and have a tail reaching over 50 cycles. Now, when looking at empirical distribution of induced gamma bursting event in visual cortex (e.g. the ones by Shapley's lab, cited in the submitted manuscript) one realizes that it is extremely unlikely to observe in vivo oscillatory bursting lasting more than 5-to-6 cycles. The authors describe their system as being in an asynchronous irregular state that can transiently resonate to boost propagation, contrasting it with Communication-through-coherence (CTC) where a "carrier wave" must be present for long times. Now, it seems to me that what the system is doing here is not producing short oscillatory transients on top of an asynchronous state, but actually triggering a collective transition from Asynchronous Irregular to a synchronous regular regime in the sender pair (if not even in the entire network). Indeed in Figure 2i one sees that a single pulse induces very quickly a long-lasting in-phase (or slightly out-of-phase) oscillation in the first two layers (1 and 2) forming the resonance pair. It is probably then the persistence of this oscillation that allow CTR from layer 2 to the outer layers to occur. It may thus be misleading to say that the mechanism here at play does not require mechanism that does not require "coherent spontaneous oscillations in the sender and receiver networks". Indeed, the propagation from layer 2 to layer 3 and above (maybe) does not require the oscillation, but without an oscillation in the sender pair this propagation would not occur. Indeed the sender module is in a real strong induced oscillatory mode as an effect of the stimulus. What this induced oscillation is doing in Figure 2i is producing an internally-generated repetition of the pulse (while in the feed-forward case of Figure 2f the pulse is repeated at the level of external input). An oscillation at the sender level is thus needed to generate internally this input pulse repetition. But is this really compatible with empirical observations, where regular, long-lasting oscillatory transients are never observed? Could the system be made working with shorter "ignition oscillatory transients"?

1b) In general, more developed comparison between this oscillation-induced CTR with models of oscillatory-transient-based CTC could be useful (the burst-mediated CTC of Palmigiano et al. 2017 e.g. relies on much shorter on average oscillatory transients than shown here in Figure 2i, however the efficiency of communication is much smaller).

2a) What if rather than a single pulse, a train of pulses is applied? Before the single pulse can be propagated an entire ignition oscillatory epoch must occur in the sender pair before propagation to outer layers starts. So what if a second pulse input arrived during this ignition oscillatory transient? Would this reset the wave, slowing down propagation or even preventing propagation of the first pulse? Would the two pulses be mixed or information about the presence of two pulses in a sequence would still retained in some way?

2b) A related question. One could think about presenting not one single pulse but a train of pulses. The some questions arise. For instance, could a temporally irregular (non periodic) train of pulses be transmitted?

2c) Or what if the frequency of the exogeneous pulse train does not match the natural frequency of the sender module (which gives the frequency of the "endogenous train of pulses" generated in Figure 2i by a single exogenous pulse). Could the network still be able to transfer trains with unnatural frequencies if delays are adjusted differently than to match the intrinsic frequency of the sender module?

2b+c) There is indeed a potential confounding factor in the study. One input pulse produce a train of pulses in the sender module. Then after a certain number of pulses propagation to layer 3 and beyond is started. Does it matter that the pulses are rhythmic, with a frequency matching the natural frequency of the sender module? Or it is just important that multiple pulses cumulate gradually lowering the excitability threshold for layer 3?

3) What if the oscillatory resonance responsible for the "ignition oscillatory transient" was generated inside the input layer 1 itself (exploiting the I-E-I loops within the sender layer, tuning e.g. the first layer to be near Hopf bifurcation) rather than by a connectivity loop involving two layers?

4a) What if the resonant pair was not exactly at the beginning but was somewhere else in the middle of the chain? Checking this would be important to understand if the enhancement of propagation is really just due to resonance at the input stage, or whether the presence of a resonant pair makes the entire chain "collectively resonant"

4b) On the same line, what if there were multiple resonant pairs, e.g. feedback everywhere (with some symmetry breaking in couplings, to keep a directionality in the chain)?

MINOR ISSUES:

Is the Fano factor really the best indicator of synchrony? It is an indicator of regulairty or irregularity of firing, but at the single neuron level. What about some index explicitly measuring synchronization at the population level?

Figure 7: it may be simpler to add a legend for the three working points, rather than having to read expicitly the caption where the symbols do not visually appear.

Figure 8: why so jittery? Are the spectra broadband so that identifying an oscillatory peak is questionable leading to noisy results? Or are the simulations performed too short? Or the fluctuations of peak frequency with the parameters are really so little smooth?

Reviewer #2: General

The contribution of this work is neither theoretical or neuroscientific, is its a computational advancement on the description of a mechanism already proposed by the group that has not been observed experimentally or understood theoretically. Without an attempt to give a mathematical insight to the phenomenon, or to contextualize it such that has biological relevance I don't think its a significant contribution.

1) The paper doesn’t explain the reader why this advancement from CTR is relevant. Where is this synchronous propagation in the brain? In which animal? In which system? Are you thinking about evoked gamma propagating through the visual hierarchy (somewhat reported in Roberts et al Neuron 2013)?

2)The study uses biological motivation to justify new choices but the final implementation seems arbitrary (only first two layers are bidirectionally connected). Figure 12 of Markov et all 2011 Cerebral Cortex shows that V1 receives more connections from V4 than V4 does from V1. I understand the idea of a hierarchy but this is a very strong simplification without an attempt to show how robust it is. A possible way forward is to have weaker degrees of bidirectionally when going through the hierarchy and show robustness. What happens if other layers are bidirectional? How weak the connections between distant layers would have to be for the phenomenon to hold?

3) The authors find that if the delays are fine tuned to match the input period then the propagation is more successful. Despite investing several figures in stating this phenomenon there is no attempt to understand this theoretically. Two E-I networks connected can be studied with linear response theory. Is the network at the onset of a hopf bifurcation? These are LIF neurons with alpha functions, a more complicated version of that was studied in Brunel & Wang 2003 and can be studied with the method developed by richardson even in the nonlinear case (Richardson 2008 Biol Cyber) and in the coupled network studied here.

4)The layers are quite small, in which is easier to generate oscillations, what happens for bigger network sizes? How is this network balanced? Do you use BE scaling or theory in any way? A purely balanced net is it inhibition stabilized, is this the case? Which is the fundamental ingredient to get resonance? How can it be understood? Is the effect robust to longer distance connecitons? and to delay heterogeneities?

Introduction

1) The introduction seems too close to the original CTR paper, and could improve its degree of argumentation. There is no mention of the plausibility of the CTR mechanism or why would it be interesting to study it. Where does this happen in the brain? What is the model system that you are trying to understand? Is there any evidence for synchronous propagation?

2) There is no evidence of uni-directional connectivity, on the contrary the data from Markov and Kennedy indicate that feedback connections are more numerous than feedforward ones, so how do we reconcile these results?

3) The CTC hypothesis, with all its pitfalls, does stem from experimental findings; whats the experimental evidence for CTR? Also, there are more transmission mechanisms than the ones described that are not even touched upon (Akam, Kullman 2010 )

4) The two limitations to CTC are not fair, one is taken from a paper that says that if CTC is true it would have to be true in small transients, and there is no evidence against that, and the other assertion, that interareal phase locking is not understood theoretically needs a bit more reading ( ermentrout kopell, battaglia brunel hansel)

Methods + Results

1)Networks are small, what happens if bigger?

2)8000 independent poisson? like compound? not clear (l.135)

3)reference for spectral entropy

4)what is a weak connection? weak compared to threshold? are these numbers close to any experimental data? which one?

5)tested the idea of connecting the first and second layer in a bidirectional manner (l.228). Where is this idea coming from? As mentioned before there is no evidence for this in the visual hierarchy, are you thinking of another system?

6)How are these network balanced? balanced networks have strong connections, and you are studying weak connections. Is there any scaling? A balanced network in the classical sense is always inhibition stabilized, is this the case?

7)Figure 3. The gamma function there is a bad fit (whats its goodness?). Given that there are simple and really fast simulations, this distribution could be better sampled.

8)Loop transmission delay (l. 249). Why this much is good needs to be understood, finding out something in a simulation is not sufficient. There is not even an intuition of why is this the case.

9)Fig 8. (b-d) color range doesn’t reveal much. Also, again, why is the frequency dependency not smooth? If there is that much variability from pixel to pixel then needs more repetitions to reveal the mean behaviour and probably larger networks.

**Have all data underlying the figures and results presented in the manuscript been provided?**

Reviewer #1: Yes

Reviewer #2: Yes

PLOS authors have the option to publish the peer review history of their article (what does this mean?). If published, this will include your full peer review and any attached files.

Reviewer #1: Yes: Demian Battaglia

Reviewer #2: No

---

## [Editor Report · Decision Letter 1]

8 Jun 2020

Dear Dr. Kumar,

We are pleased to inform you that your manuscript 'Facilitating the propagation of spiking activity in feedforward networks by including feedback' has been provisionally accepted for publication in PLOS Computational Biology.

Best regards,

Samuel J. Gershman

Deputy Editor

PLOS Computational Biology

---

## [Editor Report · Acceptance letter]

30 Jul 2020

PCOMPBIOL-D-19-01237R1 

Facilitating the propagation of spiking activity in feedforward networks by including feedback

Dear Dr Kumar,

I am pleased to inform you that your manuscript has been formally accepted for publication in PLOS Computational Biology. Your manuscript is now with our production department and you will be notified of the publication date in due course.

With kind regards,

Laura Mallard
